# Pro-active monitoring and social interventions at community level mitigate the impact of coronavirus (COVID-19) epidemic on older adults' mortality in Italy: A retrospective cohort analysis

**Giuseppe Liotta**[1]*, **Leonardo Emberti Gialloreti**[1], **Maria Cristina Marazzi**[2], **Olga Madaro**[3], **Maria Chiara Inzerilli**[4], **Margherita D'Amico**[5], **Stefano Orlando**[1], **Paola Scarcella**[1], **Elisa Terracciano**[5], **Susanna Gentili**[6], **Leonardo Palombi**[1]

**1** Biomedicine and Prevention Dept, University of Rome Tor Vergata, Rome, Italy, **2** LUMSA University, Rome, Italy, **3** Community of Sant'Egidio, 'Long Live the Elderly' Program, Rome, Italy, **4** Roma Capitale, Dipartimento promozione dei Servizi Sociali e della Salute, Roma, Italy, **5** Specialization School for Hygiene and Preventive Medicine, University of Rome Tor Vergata, Rome, Italy, **6** Doctoral School in Nursing Sciences and Public Health, University of Rome Tor Vergata, Rome, Italy

* giuseppeliotta@hotmail.com

## Abstract

### Background

The COVID-19 epidemic in Italy has severely affected people aged more than 80, especially socially isolated. Aim of this paper is to assess whether a social and health program reduced mortality associated to the epidemic.

### Methods

An observational retrospective cohort analysis of deaths recorded among >80 years in three Italian cities has been carried out to compare death rate of the general population and "Long Live the Elderly!" (LLE) program. Parametric and non-parametric tests have been performed to assess differences of means between the two populations. A multivariable analysis to assess the impact of covariates on weekly mortality has been carried out by setting up a linear mixed model.

### Results

The total number of services delivered to the LLE population (including phone calls and home visits) was 34,528, 1 every 20 day per person on average, one every 15 days during March and April. From January to April 2019, the same population received one service every 41 days on average, without differences between January-February and March-April. The January-April 2020 cumulative crude death rate was 34.8‰ (9,718 deaths out of 279,249 individuals; CI95%: 34.1–35.5) and 28.9‰ (166 deaths out of 5,727 individuals; CI95%:24.7–33.7) for the general population and the LLE sample respectively. The general population weekly death rate increased after the 11th calendar week that was not the case

**Data Availability Statement:** All relevant data are within the paper and its Supporting Information files.

**Funding:** The authors received no specific funding for this work.

**Competing interests:** The authors have declared that no competing interests exist.

among the LLE program participants (p<0.001). The Standardized Mortality Ratio was 0.83; (CI95%: 0.71–0.97). Mortality adjusted for age, gender, COVID-19 weekly incidence and prevalence of people living in nursing homes was lower in the LLE program than in the general population (p<0.001).

## Conclusions

LLE program is likely to limit mortality associated with COVID-19. Further studies are needed to establish whether it is due to the impact of social care that allows a better clients' adherence to the recommendations of physical distancing or to an improved surveillance of older adults that prevents negative outcomes associated with COVID-19.

## 1. Introduction

A disproportional rate of severe infection due to the COVID-19 epidemic was reported in Italian older adults, resulting in a very high age-specific mortality rate; in Italy, the population aged 80 years and older accounts for 7% of the total population, and more than 25% of the SARS-CoV-2 infections and about 60% of COVID-19-related deaths were reported in this population [1]. The experience of Italian older adults is like that of older adults in other European countries including Spain, the UK, France, Belgium, and Sweden where most deaths have been recorded in the population >80 years [2].

During the first months of 2020, the Italian provinces severely hit by the SARS-CoV-2 infection showed an increase in mortality of around 200% compared with that in the same period of 2019, and this figure mainly consisted of people aged >80 years [3]. Some reasons for the spread of infection in this population could be the high percentage of this population among the host of the Italian nursing homes that have been heavily affected by COVID-19, and the fragmentation of society that has led to many older adults living alone [4]. In fact, when one older person/an older couple lives alone, it could be very difficult for him/her/them to adhere to behavioural advice like the ones recommended during the COVID-19 epidemic, (e.g., stay at home) if there is no one to offer help. According to the Italian Institute of Statistics (ISTAT), more than 25% of Italian older adults claim that they cannot count on anyone for help in case of need [5]. Moreover, nursing home hosts and their carers showed many difficulties in practising social (or physical) distancing since many hosts need personal care several times per day. The lack of Individual protective devices, especially during the first phase of the epidemic spread, increased the risk of SARS-CoV-2 transmission.

Even if social fragmentation is associated with a high incidence of COVID-19 infection among Italian older adults, then social connectedness should be associated with a low mortality of COVID-19. In fact, higher mortality rate among older adults has been associated to smaller household size and higher **Long-Term Care Facility (**LTCF) bed rate [4]. The aim of this paper is to compare the mortality from COVID-19 in the general population and in a sample of older adults followed-up by the 'Long Live the Elderly!' (LLE) program, which is devoted to counteracting social isolation.

## 2. Materials and methods

This paper analyzed the trends of mortality from 1st January to 30th April 2020, and the potential determinants in the population aged above 80 years in three Italian cities: Rome, Genoa,

and Novara. These three cities were chosen because they are the only ones where the LLE program has been running, and data about mortality in the general population by age group were updated at the end of April 2020, to include the first months in which the epidemic spread in Italy. Moreover, the three cities showed different levels of epidemic spread, high, medium, and low levels in Novara, Genoa, and Rome, respectively [3], allowing comparisons in different contexts.

## 2.1 Study design and setting

The study is a retrospective cohort analysis, based on an ecological approach, carried out in three Italian cities, Genoa, Rome, and Novara.

## 2.2 Data sources

A retrospective cohort analysis has been carried out on data from two sources: the LLE electronic record database and the ISTAT database on mortality, which is freely accessible online [6].

## 2.3 Inclusion criteria

Data about mortality and services provided to the LLE program participants, gathered on a routine basis during the study period have been included in the analysis.

## 2.4 Interventions and procedures

The LLE program started in Rome (Italy) in 2004 after the 2003 heat wave [7] to meet the care needs of the older population, particularly those who had limited relationships related to age transition. It is a community-based pro-active monitoring program based on periodical phone calls and ad hoc intervention (including home visits in case of need) to meet clients' needs for care [8]. The program is implemented in agreement with the municipality, which participates especially by promoting awareness campaigns. The population included in the program has to meet two main criteria, ie living in an urban area and belonging to low socio-economic status household. All the people aged>80 living in the selected urban areas were contacted by letter and by phone. To be included they had to consent to the use of personal information for the program aims. The acceptance rate was higher than 90%, with minor differences between areas.

The frequency of phone calls is established according to the level of bio-psycho-social frailty (from one call every three months to one every two weeks). Bio-psycho-social frailty is assessed at the first contact by means of validated questionnaires and re-assessed once a year. The main objective of the program is to reach all those aged >80 years with a proactive approach and to offer them personalized and integrated social and health services according to their needs and wishes, as highlighted by the routine assessments. The aim of the LLE program is to support the client to have access to the most adequate package of services, either provided directly by the LLE program or by other available providers. Referral does not mean that the client is just addressed to a service; a continuous interaction between the provider and the LLE program is warranted. The program aims also to build a network of social relationships for the most isolated individuals to help them dealing with possible negative events, such as the death of the spouse, the development of a disease, or the worsening of physical functions. In these situations, tackling social isolation is often crucial in order to continue to live at home instead of being referred to a LTCF [8, 9].

In addition to routine activities, special interventions are activated during emergencies, such as heat waves or epidemics. All the program participants are called by phone once every two weeks (or more in case of need) to assess their physical conditions, the need for food or medicines, or to assist them in bureaucratic tasks or any other need. The aim of these actions is to allow the clients to keep the required physical distancing while avoiding social isolation. The emergency protocol started to be implemented since the first week of March until the last week of April. For this study, all the deaths that occurred from 1st January to 30th April 2020 have been recorded. In case of non-traceable individuals, news about them was gathered from their relatives or neighbours. A second wave of phone calls was performed in September to gather missing information.

## 2.5 Statistical analysis

Weekly death rates and cumulative standardized death rates were calculated. The Standardized Mortality Ratio (SMR) has been calculated according to the following formula: Number of Observed Deaths / Number of Expected Deaths. The observed deaths were the ones recorded among the LLE population, while the expected deaths were calculated applying the general population age-specific death rates to the LLE population (indirect standardization). To examine predictors of mortality, initial univariate analyses were run to identify characteristics associated with mortality at each time point. Possible determinants of mortality were assessed using Student's t-tests and Mann-Whitney U-tests for continuous variables.

For the three cities and two groups (general population and LLE population) death rates for each one of the 18 time-points were calculated. The temporal trends of the obtained death rates were then modeled using multivariable generalized regression analyses based on Repeated Measures General Linear Models (RMGLM) in order to consider the temporal correlation of the weekly incidence. In fact, mortality among people aged more than 80 could be affected by the potential mortality displacement effect due to the pre-Covid-19 epidemic (ie, a cold wave). Due to the increased prevalence of frailty in this aged population, an increase of pre-Covid-19 epidemic mortality could affect the mortality rate in the following weeks. This is why the developed RMGLM models considered the weekly incidences as within-subjects factors, the groups (general vs. LLE population) as between-subjects factors, and the mentioned variables as covariates. Therefore, the models were adjusted in each group for the interaction of baseline characteristics (age, gender, and population size). Then the weekly incidence of Covid-19 and the LTCF bed rate by city has been included in the multivariable analysis as independent ordinal variables in the model and tested the significance of each covariate as well as of the interaction terms. Interaction terms with p < 0.10 were dropped from the model. A two-sided p<0.05 was considered as statistically significant, and 95% Confidence Intervals (CIs) were reported. Analyses were conducted using SPSS, version 25.

## 2.6 Ethics statement

Regarding the LLE database, a consent form for using anonymised aggregated data for analysis was signed by the older adults when they agree to participate in the LLE program. For the present analysis only data about mortality and provided services (gathered on a routine basis) have been used.

## 3. Results

In Table 1, the population is described by age and sex according to the intervention being evaluated. The differences in terms of age and gender between the two populations are not so

**Table 1. Population by age groups and sex.**

|  |  | Age groups (years) | | | Males (%) | Females % | Individuals living in Nursing Homes (%) |
|---|---|---|---|---|---|---|---|
|  |  | 80–90 (%) | ≥90 (%) | Total |  |  |  |
| Long Live the Elderly participants | Genova | 357 (78.29) | 99 (21.71) | 456 | 34.6 | 65.4 | 6.9 |
|  | Novara | 753 (83.85) | 145(16.15) | 898 | 33.7 | 66.3 | 1.6 |
|  | Roma | 3,242 (80.25) | 845 (20.67) | 4,087 | 35.0 | 65.0 | 0.2 |
|  | **Total** | **4,352 (79.99)** | **1,089 (20.01)** | **5,441** | **34.8** | **65.2** | **1.0*** |
| Control groups | Genova | 47,380 (80.18) | 11,719 (19.82) | 59,099 | 34.7 | 65.3 | 2.7 |
|  | Novara | 6,915 (81.45) | 1,575 (18.55) | 8,490 | 35.8 | 64.2 | 4.1 |
|  | Roma | 174,978 (82.45) | 37,268 (17.55) | 212,246 | 36.1 | 63.9 | 1.3 |
|  | **Total** | **229,273 (81.94)** | **50,562 (18.06)** | **279,835** | **35.8** | **64.2** | **1.7*** |

relevant even if they must be taken into consideration in the analysis because of their impact on COVID-19 mortality [1].

The January-April 2020 cumulative crude death rate was 34.8‰ (9,718 deaths out of 279,249 individuals; CI95%:34.1–35.5) and 28.9‰ (166 deaths out of 5,727 individuals; CI95%:24.7–33.7) for the general population and the LLE sample, respectively. The LLE mean weekly mortality rate by month is lower than the general population one, except for February; however, the difference is statistically significant only in April (p = 0.006), when also the variance is similar (Levene's test and Moses test not statistically significant). The weekly mortality increase observed during March and April among the general population sample is not confirmed among the older adults participating to the LLE program (Table 2).

The death rate has been standardized (S1 Table) by age class (≤90 and >90), gender, and city to consider the differences in the three diverse epidemics. The Standardized Mortality Ratio (SMR) was 0.88 (CI95%: 0.78–1.03) for the whole period, and 0.78 (CI95%: 0.61–0.98) from March 3rd to April 26th, when the highest mortality has been recorded.

During the first four months of 2020, in the three cities chosen for this analysis, under the LLE program, 29,148 phone calls, 1,117 drug and food deliveries at home, and 2,379 home visits were made. The total number of services (also including received phone calls) was 34,528, 1 every 20 days per person on average, increased to 1 every 15 days during March and April. From January to April 2019 the same population received one service every 41 days on average, without differences between January-February and March-April.

Fig 1 shows the trend of the two populations' weekly mortality rate: it is worth of note the increase of mortality from week 10 to week 17 in the general population compared with the

**Table 2. January-April average weekly mortality rate (per 1,000) by month, according to the participation to the LLE program.**

|  |  | Mean | SD | t-test | U-Mann Withney test | Levene's test | Moses test |
|---|---|---|---|---|---|---|---|
| **January** | Gen Pop | 1.755 | 0.221 | NS | NS | p<0.001 | p>0.001 |
|  | LLE | 1.599 | 1.353 |  |  |  |  |
| **February** | Gen Pop | 1.762 | 0.313 | NS | NS | p = 0.023 | NS |
|  | LLE | 2.089 | 1.232 |  |  |  |  |
| **March** | Gen Pop | 2.811 | 1.168 | NS | NS | NS | NS |
|  | LLE | 1.973 | 1.534 |  |  |  |  |
| **April** | Gen Pop | 2.944 | 1.077 | p = 0.006 | p = 0.007 | NS | NS |
|  | LLE | 1.670 | 1.261 |  |  |  |  |
| **January-April** | Gen Pop | 2.322 | 0.974 | p = 0.025 | p = 0.026 | p = 0.042 | p>0.001 |
|  | LLE | 1.811 | 1.323 |  |  |  |  |

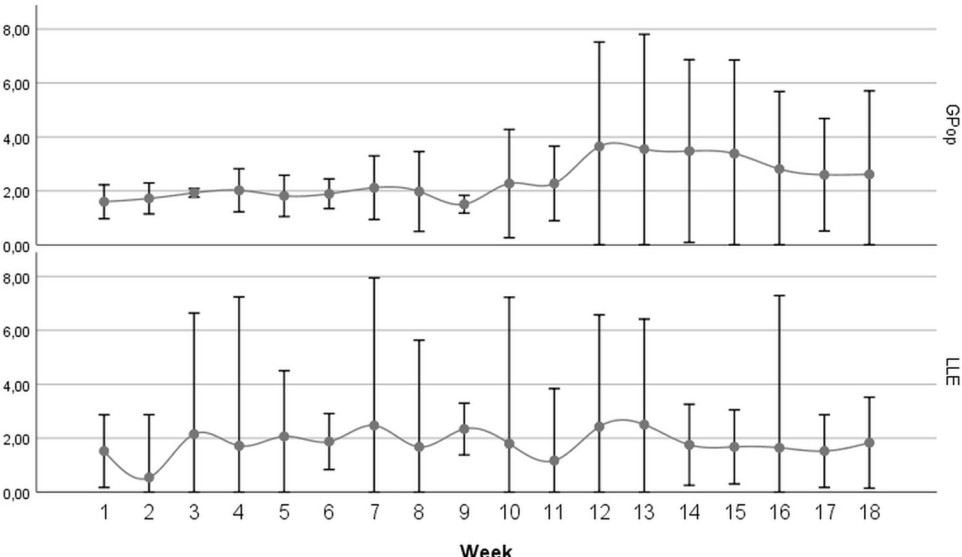

**Fig 1. Average (Rome, Genoa, Novara) weekly mortality rate (rate per 1,000 inh. and CI95%) for the general population and LLE subsamples.**

lack of increase in the same weeks observed in the LLE sample. The multivariate linear mixed model, weighted for the size of the population, and adjusted for weekly mortality rate, percentage of individuals living in nursing homes, age, and percentage of males, confirms the impact of the "Long Live the Elderly!" intervention on the mortality (Table 3, p = 0.001). The inclusion of the Intensive Care Unit (ICU) regional bed rate—a proxy of the regional health system capacity to deal with the enormous increase of ICU demand due to the COVID-19 pandemic—did not modify the results of the model.

## 4. Discussion

The paper shows the impact of COVID-19 epidemic on the older adult population in three Italian cities with different incidence of COVID-19 related mortality, according to the "exposure" to a social intervention.

The SMR synthesizes the reduction of mortality rate among the LLE participants compared with the mortality rate of the general population. This reduction was more evident from

**Table 3. Mixed linear model–fixed effect and covariances estimation[*].**

| FIXED EFFECT | Estimation | SE | p | CI95% | |
|---|---|---|---|---|---|
| | | | | Lower Limit | Upper limit |
| Intercept | 43.292 | 6.896 | <0.001 | 29.697 | 56.887 |
| **Program** (categorical, LLE = 0; GPop = 1) | 0.390 | 0.157 | 0.013 | 0.082 | 0.699 |
| COVARIANCE ESTIMATION | | Wald-Z | p | CI95% | |
| | | | | Lower Limit | Upper Limit |
| **Weekly incidence (Repeated Measures)** | | 17.652 | <0.001 | 1.075 | 1.342 |
| **Individuals living in nursing homes (%) + males (%) + age>90(%) + calendar week (interaction)** | | 1.257 | 0.209 | 0.072 | 1.630 |

• "Program" variable is included in the model as a fixed effect variable while the "weekly mortality rate", "percentage of individuals living in nursing homes", "age" and "gender", as random co-variables.

March 3rd to April 26th, and in the cities of Novara and Genoa where the epidemic was more severe than in Rome. The triple standardization (gender, age, and city) takes into consideration the different stage of epidemic in the three cities. It is worthy of note that the mortality increased by 83%, 100% and 5.4% in Novara, Genoa, and Rome, respectively [10]. The LLE program showed a positive impact on mortality mainly in cities where the negative impact of COVID-19 on mortality was more prominent.

The linear mixed model analyzed the risk of mortality growth in the control population compared with the LLE one, confirming the impact of the LLE program on the mortality containment.

There are various explanations for this phenomenon: the first one is the limited number of individuals living in nursing homes in the LLE population, especially in the Rome sub-sample. Of note, avoiding admission to a nursing home is one of the main goals of the LLE program, so it should be considered a result of the program itself. The nursing home admission rate among the citizens who participated in the program in Rome during 2019 was 0.77 per 1,000 individuals, while that for the general population in Rome was 0.97 per 1,000 individuals, with a difference of about 20% (data not published). This is probably due to the indirect effect of increasing social connectedness, promoted by the LLE program, since living alone has been associated to a more frequent recourse to LTCF [11, 12] as well as the lack of social ties [13, 14].

However, the trend of epidemic is not leaded only by LTCFs bed rate that is not likely the only factor explaining the mortality increase. The role played by the LLE program could be also associated to a reduction in hospital admission that had been already shown [7, 15]. Of note, the hospital environment was a place of transmission of SARS-CoV-2 infection, especially at the beginning of the epidemic in the last weeks of February, before the implementation of effective preventive measures on large scale in the hospital settings. The potential reduction in hospital admissions could have played a protective role in the spread of the infection among the LLE program clients.

Social isolation is associated per se to an increase of mortality [1–18] as well as pre-frailty [19]. In general, social isolation and frailty are strongly associated [20], especially in men [21], while frailty showed to be associated to death in hospitalized COVID-19 patients [22]. Hence, a third protective factor could be the proactive monitoring and the following interventions. The LLE considers the COVID-19 epidemic an emergency due to the special attention needed by the older population. The emergency protocol was activated from the beginning of March to the end of April; it includes a phone call to every client twice a month (or more when needed) to check for any requirement for practical need (food, drugs, bill payments, etc.) to allow the clients to stay safe at home as much as possible. The services delivered during the study period (home visits, food/drug home delivering) increased if compared to the same period of 2019. This shows the efforts of the LLE program to meet the increased demand for care. It is possible that the mortality containment among the program participants during the first months of the COVID 19 crisis might be generated by an increase in social capital because of the program activity along the years, as well as by the protective effect of the intervention that facilitated the clients' compliance to the general prescriptions to prevent SARS-CoV-2 contagion. For example, having someone who can control the adherence to the drug schedule and who drives the attention of the GP to a symptom might have avoided hospital admissions or access to emergency rooms, which are sometimes the starting point of dangerous vicious circles. The containment of mortality observed in the LLE population during the COVID-19 emergency is consistent with the one observed during the heat waves. The LLE emergency protocol is the same and limited the increase in mortality observed in the neighboring urban zones exposed to the same environmental stress by half [9]. The LLE program acts as a shock

**Table 4. Potential confounders.**

| |
|---|
| 1. Disparities among population level mitigation measures |
| 2. Disparities among regional health systems capacities |
| 3. Differences in terms of frequency of phone calls from friends or neighbours between the LLE and the general population |
| 4. Differences in terms of frequency of visits from friends or neighbours between the LLE and the general population |
| 5. Differences in terms of duration of the above visits/calls |
| 6. Discrepancies among action performed by the visitors (bring food or medication, drive the participants to medical appointments and/or to pharmacies to collect medications, drive respondents to shops |
| 7. Differences in illnesses among the two populations |
| 8. Differences in drugs assumption among the two populations |
| 9. Different causes of death among the two populations |
| 10. Different impact of all these factors on nursing home residents included in the study |

absorber, reducing the impact of emergencies such as the ones we analyzed. Of course, the containment of mortality in a population aged >80 years is very difficult to be observed under normal life conditions since in Italy the median age at death is 81 and 85 years for men and women, respectively. The protective action of the program was evident because of the rise of stress factors provoking an unexpected increase in mortality.

This paper presents several limitations. Firstly, the two population are not fully comparable in terms of age and gender distribution. The standardization procedure aims at overcoming the differences to allow the comparison. A limited percentage of the sample of LLE clients (<5%) was not reached by the researchers, but news about their health status have been obtained from neighbors: this might have led to a bias due to inaccurate information. However, also some information stemming from the municipality could be delayed as usually happen when dealing with official information on deaths (delay of notification). Other limitations are related to the lack of information about factors that might influence mortality, such as adherence to governmental public health measures (wearing masks, keeping distance, stay at home, etc.) that are well-known interventions to reduce mortality by decreasing transmission (Rt) [23]. Nevertheless, there are no reasons to think that these recommendations have been better accomplished by the LLE participants than the general population, except for the role played by the LLE program itself as a reinforcing factor. Therefore, we had to assume that these differences were not relevant to invalidate the results especially due to their diffusion that is likely to be similar in both populations. Further potential confounders that could bias the association between the LLE program and the containment of COVID-19 related mortality are listed in Table 4. For all these limitations, further analytical studies are warranted to confirm or question our results.

## 5. Conclusions

The analysis of the standardized death rates of people aged >80 years shows the increase in death rates observed in the general population, which is different from the one observed among the clients of the LLE program. It is likely that the intense activity of the LLE program, focused on supporting social connectedness, limited the negative impacts of the COVID-19 epidemic among older adults, especially when physical distancing is mandated. However, this conclusion should consider the limitations of the study, which are mainly related to the study design. Further studies are needed to highlight the relation between social care and mortality among people aged>80 during a pandemic crisis. In case a containment of mortality will be

confirmed it will be crucial to establish whether it is due to the impact of social care that allows a better clients' adherence to the recommendations of physical distancing or to an improved surveillance of older adults that prevents negative outcomes associated with COVID-19. Since the COVID-19 crisis will last, even if attenuated by vaccinations, the question on how to protect the older adults without increasing their social isolation—a well-known risk factor for mortality—is still open. However, the lesson learnt from the COVID-19 crisis is that each crisis calls for an increase in the attention and care provided to the frail population by community care services. The integration of health and social care at the community level is a crucial step for improving the quality of the intervention.

## Supporting information

**S1 Table. Death rate standardization (indirect procedure).**
(XLSX)

## Author Contributions

**Conceptualization:** Giuseppe Liotta, Maria Cristina Marazzi, Leonardo Palombi.

**Data curation:** Giuseppe Liotta, Olga Madaro, Maria Chiara Inzerilli, Margherita D'Amico, Paola Scarcella, Elisa Terracciano, Susanna Gentili.

**Formal analysis:** Leonardo Emberti Gialloreti, Stefano Orlando, Susanna Gentili.

**Investigation:** Maria Chiara Inzerilli, Margherita D'Amico, Stefano Orlando, Elisa Terracciano, Susanna Gentili.

**Methodology:** Leonardo Emberti Gialloreti, Stefano Orlando.

**Writing – original draft:** Giuseppe Liotta.

**Writing – review & editing:** Leonardo Emberti Gialloreti, Maria Cristina Marazzi, Stefano Orlando, Leonardo Palombi.

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
