## [Decision Letter · Decision Letter 0]

11 Aug 2021

PONE-D-21-11121

Pro-active monitoring and social interventions at community level mitigate the impact of Coronavirus (COVID-19) epidemic on older adults’ mortality in Italy: a retrospective cohort analysis

PLOS ONE

Dear Dr. liotta,

Thank you for submitting your manuscript to PLOS ONE. After careful consideration, we feel that it has merit but does not fully meet PLOS ONE’s publication criteria as it currently stands. Therefore, we invite you to submit a revised version of the manuscript that addresses the points raised during the review process.

Both reviewers feel that the study addresses an important question, and should eventually be published. However, both of them recommended that their concerns be addressed before this happens. In particular, Reviewer 1 is concerned about confounding factors that also affect COVID-19 mortality amongst seniors. These include vaccination of seniors, which reduces mortality, and the healthcare capacity being exceeded, which increases mortality. Please think about how these confounding factors can be eliminated from the data, before discussing the effectiveness of non-pharmaceutical intervention measures. Reviewer 2, on the other hand, asks for more details on the data collected.

We look forward to receiving your revised manuscript.

Kind regards,

Siew Ann Cheong, Ph.D.

Academic Editor

PLOS ONE

Journal Requirements:

Reviewers' comments:

Reviewer's Responses to Questions

**Comments to the Author**

1. Is the manuscript technically sound, and do the data support the conclusions?

Reviewer #1: Yes

Reviewer #2: No

2. Has the statistical analysis been performed appropriately and rigorously? 

Reviewer #1: Yes

Reviewer #2: No

3. Have the authors made all data underlying the findings in their manuscript fully available?

Reviewer #1: Yes

Reviewer #2: No

4. Is the manuscript presented in an intelligible fashion and written in standard English?

Reviewer #1: Yes

Reviewer #2: Yes

5. Review Comments to the Author

Reviewer #1: The authors aimed to assess if a social and health program reduced mortality associated to the COVID-19 epidemic in Italy, specifically to the older aged groups. The paper has important significance to the scientific readership in the quest to understand mitigation efforts that could reduce negative impacts to vulnerable groups. While the manuscript is well presented and written, I would like to deliver few comments that I feel the authors need to address prior to consideration into publication.

1. The intervention program (LLE) has been shown to have a positive effect to reduce mortality among old aged. Could this effect be actually confounded by national vaccination programs that have initially targeted vulnerable groups to reduce the impact on severe COVID-19 such as hospitalizations and mortalities? If this association was not tested, I suggest that the authors mention them in the limitations part or succinctly describe them briefly as potential public health implications. One suggestion is that, the infectiousness (Rt) overtime with population level mitigation measures could have, at the overall stage been reduced, due to government mitigation measures, and yet massive vaccinations targeting vulnerable groups to reduce complications or to achieve the somewhat “herd immunity” could have been likely reducing mortality rates. These interactions or confounders needs to be taken into consideration. The following literature are worthy to be cited:

• Ganasegeran, K.; Ch’ng, A.S.H.; Looi, I. What Is the Estimated COVID-19 Reproduction Number and the Proportion of the Population That Needs to Be Immunized to Achieve Herd Immunity in Malaysia? A Mathematical Epidemiology Synthesis. COVID 2021, 1, 13-19. https://doi.org/10.3390/covid1010003

• Kwok KO, McNeil EB, Tsoi MTF, Wei VWI, Wong SYS, Tang JWT. Will achieving herd immunity be a road to success to end the COVID-19 pandemic? J Infect. 2021 Jun 10:S0163-4453(21)00287-5. doi: 10.1016/j.jinf.2021.06.007.

2. The next potential confounder of deaths rates could be attributed to the fact that health systems are overwhelmed, lack of testing capacities and test sensitiveness. These should somewhat be mentioned in the introduction as potential factors to cause mortalities among the elderly, especially to those staying alone or within homes of the elderly that has the potential to yield “clusters.”

3. A crucial part that needs further elaboration here is that the description of “LLE.” The intervention was an adopted one aimed for a different purpose, “older population that faced lack of relationships associated with age transition.” It is unclear how authors could convince readers that the same program be adopted within a public health crisis such as during pandemics. One such issue I could postulate here is psychological repercussions and social isolation among the elderly that may affect some unprecedented issues, but how about the threat to the infection itself?

4. Can the authors subsection the methodology part as: study design and setting, study participants, inclusion and exclusion criteria, data sources, interventions and procedures, statistical analysis, ethics statement. This would give a more systematic reporting of the methodology part for clear readership.

5. What is CL95%? Do the authors point to confidence interval (CI)? I suggest to change to CI or define them at the beginning before the abbreviation being used.

6. Table 1 – Please include the proportion of females

7. Include the formula for SMR

8. Why choose a non-parametric approach to most of your analysis? Was your continuous data skewed?

9. The comparator was the general population, hence how could we know if LLE, specifically designed for the older aged will be effective, unless stratified by age in the comparator as well. Not very clear to me in the manuscript, but I appreciate if authors could explain.

10. Table 3 – your standard error value (SE) is quite huge. Please check.

Overall, this is a good study and worthy of publication.

Reviewer #2: This is a study of a serious problem, the absence of social and family support for elderly persons living alone. The severity of the psychosocial component was recognised by the creation of a new Ministry in England, the Ministry for Loneliness in Seniors.

Introduction, research plan and results:

In a retrospective cohort study like this the key problem is identifying which are the known confounders and then collecting data at low risk of bias to comprehensively examine these. Missing data are:

Number and content of contacts other than phone calls from the programme:

1. How were individuals selected or self-selected for the programme?

2. What were the attrition rates? What were the causes of attrition?

3. What other inputs did the participants receive?

4. How many phone calls from friends or neighbours?

5. How many phone visits from friends or neighbours?

6. How many phone calls from friends or neighbours?

7. How many phone visits from friends or neighbours?

8. Frequency and duration of the above visits/calls?

9. Did visitors bring food or medication?

10. Did visitors drive the participants to medical appointments and/or to pharmacies to collect medications?

11. Did visitors drive respondents to go shopping?

12. Did visitors perform needed house repairs? (this is a problem that sometimes causes residents to move

Illness and comorbidities:

1. What were their illnesses and comorbidities?

2. Did they have infleunza, pneumococcal or COVID-19 or other infections?

3. What medications did they take? Serious adverse effects of medications?

4. Cause of death?

The relevant items above need to be separately reported for those in nursing homes and living at home.

The proportion in nursing homes was 1.7% in the control and 1% in the intervention group (very low compared to other western countries). Did you control for this?

I mention all these considerations:

1. to ask if you have data bout them

2. To suggest that you can draw no causative conclusions from a retrospective cohort study with so many missing known confounders.

May I suggest you rewrite your study to provide as much data as you can for subsequent researchers and draw no causative conclusions. (can a phone call every three months be reasonably expected to have any effect?)

6. PLOS authors have the option to publish the peer review history of their article (what does this mean?). If published, this will include your full peer review and any attached files.

Reviewer #1: No

Reviewer #2: **Yes: **Roger E. Thomas

---

## [Author Response · Author response to Decision Letter 0]

22 Sep 2021

All the answers are included also in the rebuttal letter.

Review Comments to the Author

Reviewer #1: The authors aimed to assess if a social and health program reduced mortality associated to the COVID-19 epidemic in Italy, specifically to the older aged groups. The paper has important significance to the scientific readership in the quest to understand mitigation efforts that could reduce negative impacts to vulnerable groups. While the manuscript is well presented and written, I would like to deliver few comments that I feel the authors need to address prior to consideration into publication.

1. The intervention program (LLE) has been shown to have a positive effect to reduce mortality among old aged. Could this effect be actually confounded by national vaccination programs that have initially targeted vulnerable groups to reduce the impact on severe COVID-19 such as hospitalizations and mortalities? If this association was not tested, I suggest that the authors mention them in the limitations part or succinctly describe them briefly as potential public health implications. One suggestion is that, the infectiousness (Rt) overtime with population level mitigation measures could have, at the overall stage been reduced, due to government mitigation measures, and yet massive vaccinations targeting vulnerable groups to reduce complications or to achieve the somewhat “herd immunity” could have been likely reducing mortality rates. These interactions or confounders needs to be taken into consideration. The following literature are worthy to be cited:

• Ganasegeran, K.; Ch’ng, A.S.H.; Looi, I. What Is the Estimated COVID-19 Reproduction Number and the Proportion of the Population That Needs to Be Immunized to Achieve Herd Immunity in Malaysia? A Mathematical Epidemiology Synthesis. COVID 2021, 1, 13-19. https://doi.org/10.3390/covid1010003

• Kwok KO, McNeil EB, Tsoi MTF, Wei VWI, Wong SYS, Tang JWT. Will achieving herd immunity be a road to success to end the COVID-19 pandemic? J Infect. 2021 Jun 10:S0163-4453(21)00287-5. doi: 10.1016/j.jinf.2021.06.007.

In Italy, as well as in other EU countries, the vaccination program started on December 29, 2020, whereas data is gathered from January 1st to April 31st, 2020 (i.e. before the introduction of vaccinations). Therefore, vaccination cannot be considered a possible confounding factor. Moreover, since we are presenting a study based on an ecological approach considering the comparison between population exposed to the same mitigation interventions (wearing mask, keep distances, stay at home, etc..) it is unlikely that selection biases such to affect the results might have occurred. Nevertheless, as suggested by the reviewer, we reported the latter consideration as a study limitation

2. The next potential confounder of deaths rates could be attributed to the fact that health systems are overwhelmed, lack of testing capacities and test sensitiveness. These should somewhat be mentioned in the introduction as potential factors to cause mortalities among the elderly, especially to those staying alone or within homes of the elderly that has the potential to yield “clusters.”

This is another important point raised by the referee, and we are grateful for that. We actually presented data referring to three Italian administrative regions (Genoa–Liguria region, Novara-Piedmont, and Rome-Lazio). It is true there are differences in the capacity of the Regional Acute Health Care Systems among the Italian administrative regions especially in relation to different work loads. We tried to test the model introducing the regional Intensive Care Unit bed rate as a proxy of the capacity of the health systems to deal with an increase in the demand for acute intensive care like the one observed during the first phase of the epidemic in Italy. Result of the analysis did not substantially change, and the statistically significant result of the program was confirmed. So, we did not change the results but we presented the issue raised by the referee as one of the limitations of our study. 

 3. A crucial part that needs further elaboration here is that the description of “LLE.” The intervention was an adopted one aimed for a different purpose, “older population that faced lack of relationships associated with age transition.” It is unclear how authors could convince readers that the same program be adopted within a public health crisis such as during pandemics. One such issue I could postulate here is psychological repercussions and social isolation among the elderly that may affect some unprecedented issues, but how about the threat to the infection itself?

The reviewer is right. A more detailed description of the LLE intervention program is needed to appreciate the contribution of this program in this specific emergency. LLE was developed several years ago to deal with one of the most frequent crises faced by public health services in many Western countries for the last 20 years, ie the increase of mortality due to heat waves. During summer 2003 about 70,000 unexpected deaths have been recorded in Europe due to the heat waves. More than 90% of the deaths affected people aged more than 75, particularly when living alone. Something similar, even if not in the same size, happened during some of the following years. In the last five years before the COVID crisis (2015-2019) the LLE program showed the capacity of reducing mortality. Such a reduction was associated to social interventions aimed at supporting the older adults through a network of stable and enduring relationships. 

The hypothesis on which the present study is based is that the negative consequences of the COVID crisis, included mortality, are to some extent due to the social isolation of many older adults. Isolation that was even worsened by the need to implement social distancing as a prevention measure against SARS-CoV-2 transmission. In other words, if you are alone, without significant relationships (as it happens to an increasing segment of older adults in Western countries) you are forced to leave your house for shopping or for getting drugs, exposing yourself to risk of transmission. If you have someone who can do these tasks for you, decreases such risky behaviour. Moreover, reduced contacts with your GP, as it happened during the crisis, can worsen your physical conditions, provoking hospitalizations and deaths even if not directly related to COVID-19. Of course, to increase its effectiveness it is much better if such a program is adopted as a standard of care all along the whole year than just during emergency, Nevertheless, many Italian municipalities launched similar programs to support older adults during the emergency because of the alarming number of older adults confined at home without any help.

All these considerations have been now synthesized and presented in the discussion. We believe that the remarks of the reviewer helped us to better describe and discuss the reasons of the impact of the intervention. 

4. Can the authors subsection the methodology part as: study design and setting, study participants, inclusion and exclusion criteria, data sources, interventions and procedures, statistical analysis, ethics statement. This would give a more systematic reporting of the methodology part for clear readership.

The methods section has been rearranged according to the referee’s suggestions

5. What is CL95%? Do the authors point to confidence interval (CI)? I suggest to change to CI or define them at the beginning before the abbreviation being used.

Thank you for highlighting the typo. CL has been now corrected in CI

6. Table 1 – Please include the proportion of females

Proportion of females has been added to the table

7. Include the formula for SMR

Formula for SMR has been added to the Supplementary Table 1

8. Why choose a non-parametric approach to most of your analysis? Was your continuous data skewed?

The present study is an ecological one; several analysed data have not been collected by us but were retrieved from different published sources. Considering the whole Italian population of older adults, our sample was relatively small and skewed in the direction of cities (rather than countryside). In some cases, outliers were present. These outliers could not be legitimately removed as they were a genuine part of the distribution for the study area. For all these reasons we chose the more conservative nonparametric approach. 

9. The comparator was the general population, hence how could we know if LLE, specifically designed for the older aged will be effective, unless stratified by age in the comparator as well. Not very clear to me in the manuscript, but I appreciate if authors could explain.

The comparator was always the general population aged > 80. The reviewer is right, this was not clear. We therefore added a clarification sentence in the paper.

10. Table 3 – your standard error value (SE) is quite huge. Please check.

Thank you, we double checked the SEs. They are correct, even if they are large due to the factors highlighted under bullet point 8.

Overall, this is a good study and worthy of publication.

Thank you very much for your very useful contribution

 

Reviewer #2: This is a study of a serious problem, the absence of social and family support for elderly persons living alone. The severity of the psychosocial component was recognised by the creation of a new Ministry in England, the Ministry for Loneliness in Seniors.

Introduction, research plan and results:

In a retrospective cohort study like this the key problem is identifying which are the known confounders and then collecting data at low risk of bias to comprehensively examine these. Missing data are:

Number and content of contacts other than phone calls from the programme:

In the manuscript we briefly presented the intervention delivered by the LLE program during the study period: 

“During the first four months of 2020, in the three cities chosen for this analysis, under the LLE program, 29,148 phone calls, 1,117 drug and food deliveries at home, and 2,379 home visits were made. The total number of services (including also phone calls received) was 34,528, 1 every 20 days per person on average, increased to 1 every 15 days during March and April. From January to April 2019 the same population received one service every 41 days on average, without differences between January-February and March-April”.

However, the reviewer is right in underlining that this was not taken up in the discussion section. So we considered to do that in order to make more understandable the reasons why we associated the LLE program with a possible containment of mortality. The referee pointed out a lack of a clear link between the LLE program and mortality reduction. We acknowledge that due to the characteristics of such an ecological study, some information to prove the link is still lacking. So, we decided to attenuate our conclusion on these aspects.

1. How were individuals selected or self-selected for the programme?

The programme contacts all people aged>80 leaving in the areas selected for the intervention. The main selection criteria was living in a urban area with low socio-economic status. This is relevant because it is known that low socio-economic status is a risk factor for mortality. In this analysis a proxy of socio-economic status - the mean value of owned houses - was included; it did not show any statistically significant association with COVID 19 mortality. In any case, information about selection of urban areas and population has been now added to the methods section

2. What were the attrition rates? What were the causes of attrition?

The attrition rate during the four months of the study was less than 5%, even if information related to survival have been gathered also in this case, as reported in the limitation paragraph at the end of the paper. Main cause of attrition was moving to another location. The program acceptance rate is on average higher than 90%, and during the COVID crisis some clients who initially refused to be included into the program turned later to it because of the difficulties related to the COVID crisis. 

Regarding some of the questions from 3 to 12, due to the characteristics of this study we cannot provide information about aspects not related to the LLE program operators or program volunteers. We have information about the consistency of the social network of relationship of the people included in the LLE program, as these pieces of information were gathered through the periodical assessment of frailty. But we cannot compare this information with the one of the general population (not included in the LLE program), as in this case these parameters were not available.

However, we have no evidence that allows us to state that the LLE program clients received a different input by friends/neighbours than the general population (apart from the program itself). We assume that on a similar base of social contacts the LLE program represents a booster with a relevant implication in the quality of life, resulting eventually also in a containment of mortality.

3. What other inputs did the participants receive?

4. How many phone calls from friends or neighbours?

5. How many phone visits from friends or neighbours?

6. How many phone calls from friends or neighbours?

7. How many phone visits from friends or neighbours?

8. Frequency and duration of the above visits/calls?

9. Did visitors bring food or medication?

10. Did visitors drive the participants to medical appointments and/or to pharmacies to collect medications?

11. Did visitors drive respondents to go shopping?

12. Did visitors perform needed house repairs? (this is a problem that sometimes causes residents to move

Illness and comorbidities:

1. What were their illnesses and comorbidities?

2. Did they have influenzas, pneumococcal or COVID-19 or other infections?

3. What medications did they take? Serious adverse effects of medications?

4. Cause of death?

Due to the study design this information is available for the LLE clients, but not for the general population; so we decided to exclude it from the analysis. However, we do not have reasons to think there were relevant differences between the LLE population and the general population about these issues.

The relevant items above need to be separately reported for those in nursing homes and living at home.

Unfortunately, due to the study design this information is not available

The proportion in nursing homes was 1.7% in the control and 1% in the intervention group (very low compared to other western countries). Did you control for this?

The percentage of people leaving in nursing homes has been included into the model as a co-factor

I mention all these considerations:

1. to ask if you have data bout them

2. To suggest that you can draw no causative conclusions from a retrospective cohort study with so many missing known confounders.

May I suggest you rewrite your study to provide as much data as you can for subsequent researchers and draw no causative conclusions. (can a phone call every three months be reasonably expected to have any effect?)

According to the suggestions of the reviewer, we tried now to include in the manuscript as much information as possible. However, as pointed out previously many pieces of information were not available. We agree with the reviewer that some conclusions must be softened, and we did it in this revised version.

Nevertheless, we would like to point out that the LLE program is not just based on a phone call every three months; it is rather a complex community care model which aims to increase the social capital of the population aged>80, representing the potential vector of a number of preventive interventions (like – just to mention the last example- the COVID-19 vaccination). Social isolation is a well-known risk factor for mortality, especially among the older adults population. It is possible that an intervention that counteracts social isolation can contain mortality in this population. Our data point in this direction, even if – of course – due to the many limitations of this study they cannot be considered conclusive.

---

## [Decision Letter · Decision Letter 1]

11 Oct 2021

PONE-D-21-11121R1Pro-active monitoring and social interventions at community level mitigate the impact of Coronavirus (COVID-19) epidemic on older adults’ mortality in Italy: a retrospective cohort analysisPLOS ONE

Dear Dr. liotta,

Thank you for submitting your manuscript to PLOS ONE. After careful consideration, we feel that it has merit but does not fully meet PLOS ONE’s publication criteria as it currently stands. Therefore, we invite you to submit a revised version of the manuscript that addresses the points raised during the review process.

After the first revision, Reviewer 2 remained dissatisfied with the manuscript. Reviewer 2 felt that compelling conclusions cannot be made of the study, because of the large number of known and unknown confounders. Reviewer 2 requests that the authors include a table of all confounding factors, including those suggested by the two reviewers, how these can affect the conclusions, and how data on them may be collected in future studies that can help resolve how important they might actually be.

We look forward to receiving your revised manuscript.

Kind regards,

Siew Ann Cheong, Ph.D.

Academic Editor

PLOS ONE

Reviewers' comments:

Reviewer's Responses to Questions

**Comments to the Author**

1. If the authors have adequately addressed your comments raised in a previous round of review and you feel that this manuscript is now acceptable for publication, you may indicate that here to bypass the “Comments to the Author” section, enter your conflict of interest statement in the “Confidential to Editor” section, and submit your "Accept" recommendation.

Reviewer #1: (No Response)

Reviewer #2: (No Response)

2. Is the manuscript technically sound, and do the data support the conclusions?

Reviewer #1: Yes

Reviewer #2: Partly

3. Has the statistical analysis been performed appropriately and rigorously? 

Reviewer #1: Yes

Reviewer #2: Yes

4. Have the authors made all data underlying the findings in their manuscript fully available?

Reviewer #1: Yes

Reviewer #2: Yes

5. Is the manuscript presented in an intelligible fashion and written in standard English?

Reviewer #1: Yes

Reviewer #2: Yes

6. Review Comments to the Author

Reviewer #1: Thank you for your revision. While I agree to almost all author responses and revisions, there are still two minor comments that I think authors need to address and provide appropriate conceptual evidence before the manuscript qualifies for publication.

The authors have mentioned in response to comment 1 that mitigation and containment strategies by the government could have somewhat influenced the number of cases and mortality. Authors responded in the last sentence “Nevertheless, as suggested by the reviewer, we reported the latter consideration as a study limitation.”

Authors added the following sentence in the limitations part “adherence to governmental public health measures (wearing masks, keeping distance, stay at home, etc.; diseases affecting the studied population or the causes of death).” These measures were known to affect the infectiousness (Rt) overtime, thereby determining cases escalation or decrease over time, yet may cause the mortality rates to increase or decrease overtime if health systems were overwhelmed or cases were undetected. As authors agreed to the suggestion and noted in the limitation, it needs to be justified with evidence as conceptualized in the reviewer’s comments. The following evidence needs to be cited to in accordance to authors acceptance of the proposed work and need to be corroborated based on the above argument:

Ganasegeran, K.; Ch’ng, A.S.H.; Looi, I. What Is the Estimated COVID-19 Reproduction Number and the Proportion of the Population That Needs to Be Immunized to Achieve Herd Immunity in Malaysia? A Mathematical Epidemiology Synthesis. COVID 2021, 1, 13-19. https://doi.org/10.3390/covid1010003

Second, the SMR formula needs to be incorporated under methods part as an operational definition.

Once these revisions have been made, I have no further objections for the paper to be published.

Reviewer #2: Thank you for your detailed replies to the reviewers. Supporting lonely isolated older people at risk is a key problem facing Western societies, Japan...

As has been stressed, you cannot make conclusions that the programme had an effect because of the large numbers of known and unknown confounders. I would like you to make a table listing all the confounders and items of data you were not able to collect listed by the two reviewers to illustrate to future researchers the data that they need to collect. This is not a criticism of your work, but important that as part of your scientific contribution your guide future researchers to identify methods to collect as much information on confounders as possible. Both reviewers listed large numbers of items of possible contacts that you were not able to assess

7. PLOS authors have the option to publish the peer review history of their article (what does this mean?). If published, this will include your full peer review and any attached files.

Reviewer #1: No

Reviewer #2: **Yes: **Roger E. Thomas

---

## [Author Response · Author response to Decision Letter 1]

2 Nov 2021

Dear Reviewers

please find below the answers to your suggestions that we included fully in the final manuscript. On behalf of the group of authors, I want to thank you for your contribution to improve the paper

with my best regards

Giuseppe Liotta

Reviewer #1: Thank you for your revision. While I agree to almost all author responses and revisions, there are still two minor comments that I think authors need to address and provide appropriate conceptual evidence before the manuscript qualifies for publication.

Thank you for your contribution to the improvement of the paper. Please find the answers to your comment by points

The authors have mentioned in response to comment 1 that mitigation and containment strategies by the government could have somewhat influenced the number of cases and mortality. Authors responded in the last sentence “Nevertheless, as suggested by the reviewer, we reported the latter consideration as a study limitation.”

Authors added the following sentence in the limitations part “adherence to governmental public health measures (wearing masks, keeping distance, stay at home, etc.; diseases affecting the studied population or the causes of death).” These measures were known to affect the infectiousness (Rt) overtime, thereby determining cases escalation or decrease over time, yet may cause the mortality rates to increase or decrease overtime if health systems were overwhelmed or cases were undetected. As authors agreed to the suggestion and noted in the limitation, it needs to be justified with evidence as conceptualized in the reviewer’s comments. The following evidence needs to be cited to in accordance to authors acceptance of the proposed work and need to be corroborated based on the above argument:

Ganasegeran, K.; Ch’ng, A.S.H.; Looi, I. What Is the Estimated COVID-19 Reproduction Number and the Proportion of the Population That Needs to Be Immunized to Achieve Herd Immunity in Malaysia? A Mathematical Epidemiology Synthesis. COVID 2021, 1, 13-19. https://doi.org/10.3390/covid1010003

Please see the discussion section, we have tried to report in the best possible way the considerations of the reviewer

Second, the SMR formula needs to be incorporated under methods part as an operational definition.

We added the SMR formula to the methods

Once these revisions have been made, I have no further objections for the paper to be published.

Reviewer #2: 

Thank you for your detailed replies to the reviewers. Supporting lonely isolated older people at risk is a key problem facing Western societies, Japan...

As has been stressed, you cannot make conclusions that the programme had an effect because of the large numbers of known and unknown confounders. I would like you to make a table listing all the confounders and items of data you were not able to collect listed by the two reviewers to illustrate to future researchers the data that they need to collect. This is not a criticism of your work, but important that as part of your scientific contribution your guide future researchers to identify methods to collect as much information on confounders as possible. Both reviewers listed large numbers of items of possible contacts that you were not able to assess

We added a table with the potential confounding factors w cannot assess at the end of the discussion section. thank you very much for your contribution

---

## [Decision Letter · Decision Letter 2]

6 Dec 2021

Pro-active monitoring and social interventions at community level mitigate the impact of Coronavirus (COVID-19) epidemic on older adults’ mortality in Italy: a retrospective cohort analysis

PONE-D-21-11121R2

Dear Dr. liotta,

We’re pleased to inform you that your manuscript has been judged scientifically suitable for publication and will be formally accepted for publication once it meets all outstanding technical requirements.

Kind regards,

Siew Ann Cheong, Ph.D.

Academic Editor

PLOS ONE

Additional Editor Comments (optional):

Reviewers' comments:

Reviewer's Responses to Questions

**Comments to the Author**

1. If the authors have adequately addressed your comments raised in a previous round of review and you feel that this manuscript is now acceptable for publication, you may indicate that here to bypass the “Comments to the Author” section, enter your conflict of interest statement in the “Confidential to Editor” section, and submit your "Accept" recommendation.

Reviewer #1: All comments have been addressed

Reviewer #2: All comments have been addressed

2. Is the manuscript technically sound, and do the data support the conclusions?

Reviewer #1: Yes

Reviewer #2: Yes

3. Has the statistical analysis been performed appropriately and rigorously? 

Reviewer #1: Yes

Reviewer #2: Yes

4. Have the authors made all data underlying the findings in their manuscript fully available?

Reviewer #1: Yes

Reviewer #2: Yes

5. Is the manuscript presented in an intelligible fashion and written in standard English?

Reviewer #1: Yes

Reviewer #2: Yes

6. Review Comments to the Author

Reviewer #1: The latest version of the revised manuscript is acceptable. Authors have addressed my suggestions well. Thank you.

Reviewer #2: Thanks to the authors for their careful responses to all the reviewers' suggestions. The table of potential confounders is very helpful.

This is an interesting study and the authors are to be commended. The manuscript is carefully written and the conclusions are now appropriately guarded.

7. PLOS authors have the option to publish the peer review history of their article (what does this mean?). If published, this will include your full peer review and any attached files.

Reviewer #1: No

Reviewer #2: **Yes: **Roger E. Thomas

---

## [Editor Report · Acceptance letter]

13 Dec 2021

PONE-D-21-11121R2 

Pro-active monitoring and social interventions at community level mitigate the impact of Coronavirus (COVID-19) epidemic on older adults’ mortality in Italy: a retrospective cohort analysis 

Dear Dr. liotta:

I'm pleased to inform you that your manuscript has been deemed suitable for publication in PLOS ONE. Congratulations! Your manuscript is now with our production department. 

Kind regards, 

on behalf of

Dr. Siew Ann Cheong 

Academic Editor

PLOS ONE